# Non-flipping DNA glycosylase AlkD scans DNA without formation of a stable interrogation complex

Arash Ahmadi [1], Katharina Till[2,3], Paul Hoff Backe [1,4], Pernille Blicher[1], Robin Diekmann[3], Mark Schüttpelz [3], Kyrre Glette [5], Jim Tørresen [5], Magnar Bjørås[4,6], Alexander D. Rowe[1,7] & Bjørn Dalhus [1,4✉]

The multi-step base excision repair (BER) pathway is initiated by a set of enzymes, known as DNA glycosylases, able to scan DNA and detect modified bases among a vast number of normal bases. While DNA glycosylases in the BER pathway generally bend the DNA and flip damaged bases into lesion specific pockets, the HEAT-like repeat DNA glycosylase AlkD detects and excises bases without sequestering the base from the DNA helix. We show by single-molecule tracking experiments that AlkD scans DNA without forming a stable interrogation complex. This contrasts with previously studied repair enzymes that need to flip bases into lesion-recognition pockets and form stable interrogation complexes. Moreover, we show by design of a loss-of-function mutant that the bimodality in scanning observed for the structural homologue AlkF is due to a key structural differentiator between AlkD and AlkF; a positively charged β-hairpin able to protrude into the major groove of DNA.

[1] Department of Medical Biochemistry, Institute for Clinical Medicine, University of Oslo, Oslo, Norway. [2] FOM Institute AMOLF, Science Park 104, Amsterdam, The Netherlands. [3] Biomolecular Photonics, Department of Physics, University of Bielefeld, Bielefeld, Germany. [4] Department of Microbiology, Oslo University Hospital HF, Rikshospitalet and University of Oslo, Oslo, Norway. [5] Department of Informatics, University of Oslo, Oslo, Norway. [6] Department of Clinical and Molecular Medicine, Faculty of Medicine and Health Sciences, Norwegian University of Science and Technology (NTNU), Trondheim, Norway. [7] Department of Newborn Screening, Division of Child and Adolescent Medicine, Oslo University Hospital, Oslo, Norway. ✉email: bjorn.dalhus@medisin.uio.no

The base-excision repair (BER) pathway handles aberrant nucleotides, such as alkylated or oxidized bases, in genomic DNA. The pathway consists of several enzymes working together in consecutive steps to remove and replace the modified and often mutagenic base—starting with DNA glycosylases in the first step, which produce an apurinic/apyrimidinic (AP) site[1]. DNA glycosylases can be classified into the structural super-families helix–hairpin–helix (HhH), helix-two-turn-helix (H2TH), and HEAT-like repeat (HLR), as well as the uracil-DNA glyco-sylase (UNG) and alkyladenine-DNA glycosylase (AAG) families[1,2]. Four of these are present in human cells, while the HEAT-like repeat family has only been detected in bacteria, archaea, and some simple uni- and multicellular eukaryotes[3–6]. Atomic-resolution structures of representative members of each family have been determined, both with and without DNA. A distinctive feature of the HhH, H2TH, UNG-, and AAG-families of DNA glycosylases is a characteristic flipping of DNA bases into the lesion-recognition pocket for base interrogation, damage recognition, and excision[2].

The discovery of the DNA-bound complex of the HLR DNA glycosylase AlkD revealed the first example of a DNA glycosylase whose activity does not depend on base-flipping or base-probing using a wedge residue as an essential part of DNA base inter-rogation, damage recognition, and excision[4,6,7]. AlkD removes the lesions N3-yatakemycinyl-2'-deoxyadenosine (d3yA), N3-methyl-2'-deoxyadenosine (d3mA), and N7-methyl-2'-deox-yguanosine (d7mG) from DNA[5–7], and in the case of the bulky d3yA adduct, damage recognition takes place by interaction with both the phosphoribose backbone and the d3yA compound[7,8]. The smaller and intrinsically labile d3mA and d7mG bases are conversely believed to be removed by hydrolysis due to stabili-zation of the increased positive charge on the deoxyribose back-bone through electrostatic and CH–$\pi$ interactions[7,9]. The structure of AlkF, a distant HLR homolog of AlkD by sequence but a close structural homolog, also reveals a protein lacking both

a nucleobase binding pocket for base flipping and an intercalating residue for base probing[10]. This makes both AlkD and AlkF relevant candidates to investigate the effect of none-base flipping on the overall dynamic process of scanning. However, in contrast to AlkD, AlkF contains a characteristic additional $\beta$-hairpin carrying several positively charged residues believed to protrude into the major groove of DNA. The protein has an affinity for a variety of branched DNA substrates, but no associated catalytic activity has so far been detected[10]. Whether such branched sub-strates are biologically relevant substrates for AlkF remains to be shown.

In this study, using a single-molecule approach[11] fluorescently labeled AlkD and AlkF proteins were tracked, while scanning a 12 kbp λ-DNA elongated in a linear form. Analysis of trajectories shows that scanning of AlkD resembles a homogenous random walk without the explicit modality caused by the formation of a stable interrogation complex, which contrasts previously reported multimode scanning by other DNA repair proteins[11]. This result resonates well with the unique lack of base flipping in AlkD. In contrast to AlkD, and more similar to other glycosylases, AlkF displays an explicit bimodality in DNA scanning. However, we show that unlike other glycosylases this bimodality in scanning is not caused by base flipping or base probing but is due to a higher variation in the energy barrier for scanning, which is facilitated by interaction between the positively charged $\beta$-loop and DNA. Further, we introduce the concept of redundancy and efficiency of scanning, and by comparing six different proteins in this context, we find an inverse correlation between redundancy and efficiency of scanning, reflecting the different roles and/or structures of these proteins.

## Results and discussion

To observe and characterize the scanning of proteins along a linear track of DNA, a 12 kbp λ-DNA was elongated between an anchoring point on a coverslip surface and an optically trapped polystyrene bead (Fig. 1). After injection of fluorescently labeled proteins to the point of observation, trajectories of single proteins scanning along the DNA (Fig. 1, lower panel and Supplementary Movies 1–3) were recorded using highly inclined and laminated optical sheet (HILO) fluorescence microscopy[12,13]. From these recorded trajectories, we investigated and compared the DNA-scanning behavior of AlkD (Fig. 2a), AlkF (Fig. 2b), and AlkF-Δpos (Fig. 2c), a mutant of AlkF, where three positively charged residues in the major groove pro-truding β-hairpin have been replaced with neutral residues. In order to detect statistically significant variations in the diffusion rate of the proteins under study, the instantaneous diffusion rates were calcu-lated over all trajectories, and their distributions (red density curves, Fig. 2d) were analyzed and compared with those of corresponding single-mode simulated random walks (solid black density curves, Fig. 2d).

Deviations of a protein's diffusion rate distribution from that of a simulated single-mode random walk are due to variations of the energy barrier that proteins face while stepping from one base to the adjacent. Different ranges of these energy barriers represent distinct modes of scanning. According to previous theoretical and experimental reports, the energy barrier of an uninterrupted and efficient helical sliding is within the range of 0.5–2 $k_BT$[11,14–21]. The more stable the interrogation complex is, the higher the energy barrier for stepping, with $E_a > 2\,k_BT$, until a stable recognition complex is formed for $E_a > 5\,k_BT$. Assuming that the stepping process can be considered as a kinetic reaction, a rate constant is calculated from the observed diffusion rate of scan-ning. From the rate constant, the energy barrier of base-to-base stepping ($E_a$) can be calculated for protein trajectories. Based on this concept, trajectories were segmented into high ($E_a > 2\,k_BT$)

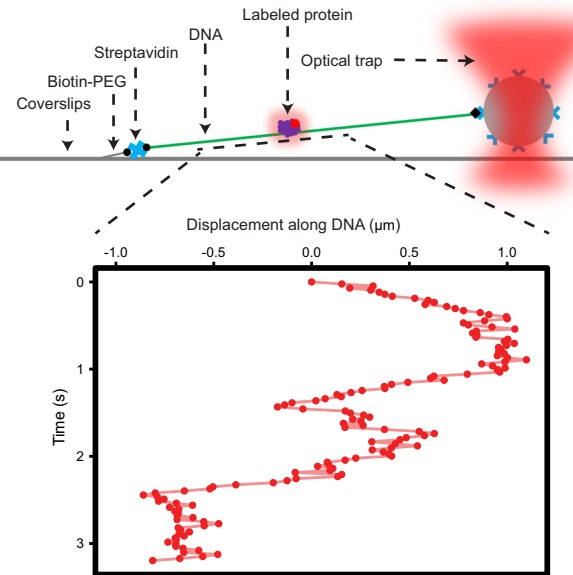

**Fig. 1 DNA-scanning experimental setup.** Using a streptavidin linker, a biotin-tagged DNA is attached to a biotin-tagged PEG molecule that is bound to the surface of a coverslip. At the other end, the anchored DNA is attached to a polystyrene bead using an anti-digoxigenin and digoxigenin link. DNA is elongated by trapping and moving the attached polystyrene bead using an optical trap. Movements of fluorescently labeled proteins along the linear track of DNA is recorded, as exemplified in the time-position plot in the lower panel.

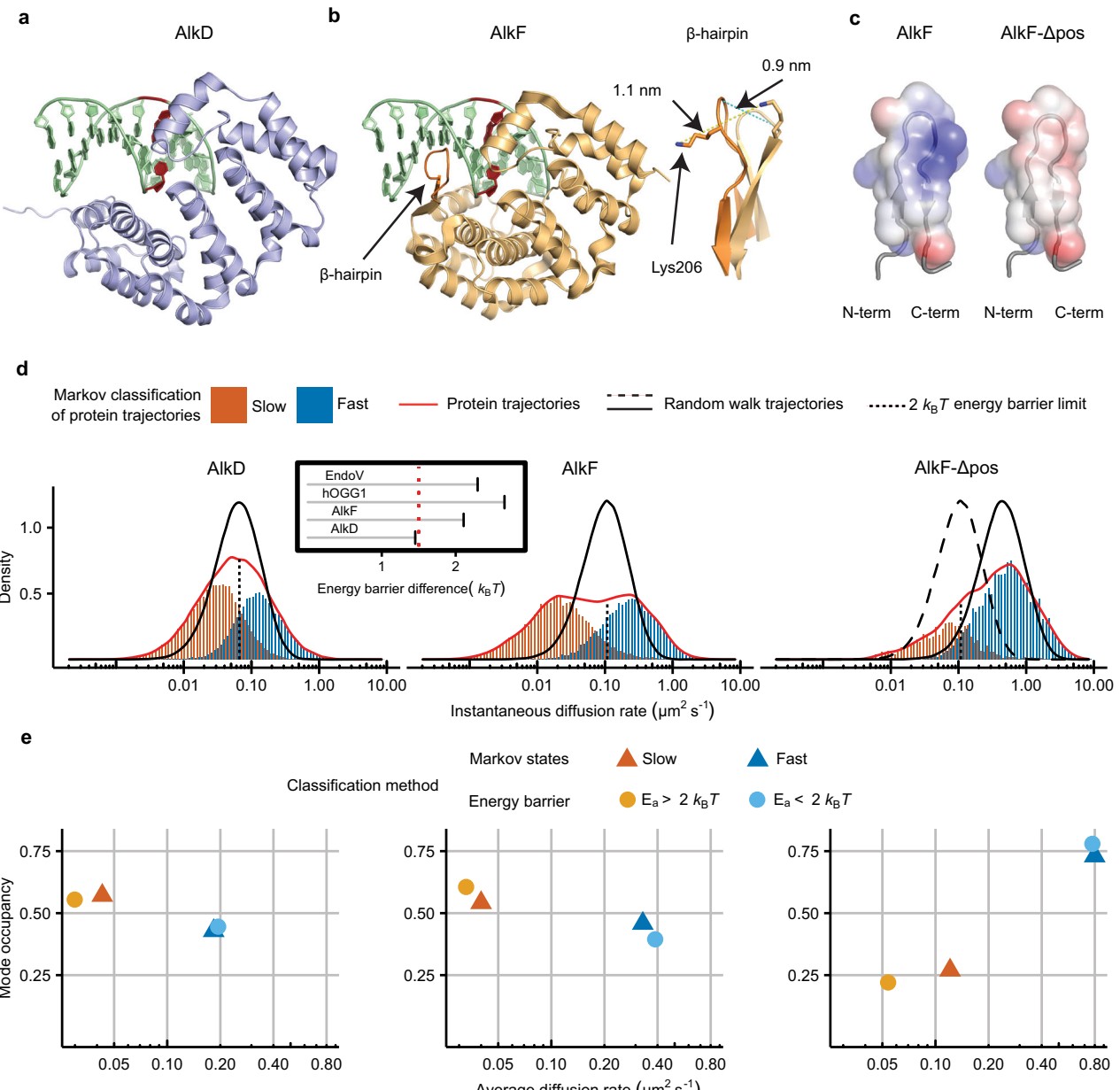

**Fig. 2 DNA-scanning analysis of AlkD, AlkF, and AlkF-Δpos. a** Crystal structure of *B. cereus* AlkD binding to DNA with 3-deaza-3-methyladenine (red base). PDB code: 3JX7[4]. **b** Model of AlkF–DNA using the crystal structure of *B. cereus* AlkF without DNA (PDB code: 3ZBO[10]) superposed onto AlkD bound to DNA. The β-hairpin of AlkF (dark orange; arrow) is protruding into the DNA major groove; Inset: a close-up view of the β-hairpin showing the flexibility of the loop structure. **c** a close-up view of the wild-type AlkF and AlkF-Δpos mutant, respectively, showing the difference in surface charge (blue/red correspond to $3/-3\ k_BTe^{-1}$). **d** Overall instantaneous diffusion rate distributions (red curves) and classification of trajectories using a hidden Markov model, with the blue and orange histograms showing the individual distributions for the fast and slow modes of the scanning, respectively. These modes represent fractions of the movement with relatively higher and lower diffusion rates for each protein independently and do not necessarily correlate with those of other proteins. The black solid lines show the density distributions for the instantaneous diffusion rates for corresponding simulated random walks, and with the dashed line in the AlkF-Δpos panel showing also the curve for wild-type AlkF for comparison. Vertical dotted lines show the border between high- and low-energy barrier modes for the kinetic model, defined by the $2\ k_BT$ energy barrier limit; inset: energy barrier difference between the two modes of HMM analysis for hOGG1, EndoV, AlkD, and AlkF. The red dotted line shows the range of energy barrier variation of helical sliding (the data for hOGG1 and EndoV from Ahmadi et al.[11]). **e** Comparison of average diffusion rate (x axis) and mode occupancy (y axis) for AlkD, AlkF, and AlkF-Δpos calculated using the kinetic energy barrier method (circles) and the independent hidden Markov model (triangles).

and low ($E_a < 2\ k_BT$) energy barriers which is indicated by the dotted vertical lines in Fig. 2d. In addition, to investigate the existence of different modes of diffusion using an independent and purely statistical approach, all frames in all trajectories were binned into either slow or fast modes (colored histograms, Fig. 2d) by applying the variational Bayes hidden Markov model

(HMM) implemented in the single-particle tracking software vbSPT[22]. Time series of protein trajectories are the observable component of a Markov chain, and parameters such as average diffusion rate, occupancy of hidden diffusion states, and transition probabilities between states are estimated by variational Bayesian treatment of the HMM.

The overall diffusion rate distribution of AlkD resembles a monomodal random walk (Fig. 2d, red and black curves); this close resemblance suggests that the sliding of AlkD along DNA is a rather smooth and homogenous movement. This contrasts with our recent findings with the DNA glycosylase OGG1 and the endonuclease EndoV[11], where we observe a clear bimodal DNA scanning with two explicit peaks in the diffusion rate distribution. The two peaks show more than one order of magnitude difference in the diffusion rate, corresponding to an energy barrier difference of $2.66\,k_BT$ and $2.30\,k_BT$ for a two-mode Markov classification of hOGG1 and EndoV, respectively. This difference in the energy barrier of the two modes of OGG1 and EndoV exceeds the $1.5\,k_BT$ energy barrier range for normal helical sliding[11,14–21] by a 77% and 53%, respectively (Fig. 2d, inset), which is compatible with the formation of a stable interrogation complex due to base flipping and/or base probing using base-flipping pockets and wedge motifs present in these enzymes[23–26]. The absence of an explicit bimodality in the overall diffusion rate distribution of AlkD (Fig. 2d, red curve) may very well reflect the absence of such mechanistic functionality to form a stable interrogation complex by AlkD. This resonates well with the fact that AlkD does not have a wedge motif/residue or base lesion pocket, but instead removes bases without base flipping[4,7]. We conclude that the lack of structural motifs facilitating base flipping, and therefore lack of a stable interrogation complex, favors an overall monomodal scanning for the AlkD glycosylase. However, classification of the trajectories into two modes using HMM analysis (Fig. 2d, orange and blue histograms) indicates the existence of an intermediate state in DNA scanning within an apparent single-mode distribution of the overall diffusion rate of AlkD (Fig. 2d, red curve). Unlike interrogation complexes formed due to base flipping, this intermediate state is not stable enough to create an explicit bimodality in the overall diffusion rate distribution. Moreover, the average diffusion rates of the slow and fast modes of AlkD scanning as classified by the HMM analysis differs by a factor of 4.3, which corresponds to $1.46\,k_BT$ in terms of energy barrier, which is within the $1.5\,k_BT$ range for helical sliding (Fig. 2d, inset). This is in contrast with the formation of stable interrogation complexes as seen for hOGG1 and EndoV with energy barrier differences between the two modes well beyond the range of helical sliding. However, the existence of these two modes for AlkD indicates that although stable interrogation complexes are not formed by this protein, intermediate states with considerably lower variation in energy barrier than that of a stable interrogation complex could exist.

Indeed, the absence of a stable interrogation complex and an explicit bimodality in the diffusion rate distribution of AlkD is consistent with a recent simulation study[27], where it is suggested that in contrast to glycosylases that flip the base, there is no distinct interrogation complex in the transition between the search (freely scanning) and excision (stationary) complex for AlkD. However, in the same study, it is suggested that due to conformational shifts observed in the AlkD–DNA contact, more subtle intermediate states could exist, which is consistent with the result of our HMM analysis showing a small difference between the energy barriers of the two scanning modes. In addition, a recent experimental study by Peng et al.[28] also showed that AlkD employs two distinct modes in the scanning of short DNA segments, which is consistent with the results of our HMM analysis for AlkD. We calculate an average diffusion rate of $0.043\,\mu m^2\,s^{-1}$ for the slow diffusion mode of AlkD, which corresponds well with the range of $0.007–0.058\,\mu m^2\,s^{-1}$ for the slow mode found by Peng et al. However, the average diffusion rate of the fast mode is around fivefold higher in the Peng study compared to the fast mode in our data, with 0.92 and $0.18\,\mu m^2\,s^{-1}$, respectively. The discrepancy between these two investigations could possibly be traced back to substantially different experimental setups or methodological approaches for the calculation of the respective diffusion rates. In the Peng study, short DNA oligos of only 21–99 base pairs are used, and the diffusion rate is calculated from an estimation of the residence time of protein per base pair, based on FRET signals between labeled AlkD and DNA, and with microsecond temporal resolution. In this study, we directly observe AlkD scanning up to a few thousand base pairs per single binding event, and the diffusion rate is calculated by tracking the stepping of the protein with millisecond temporal resolution. It could be that confining the movement of AlkD to <100 bp oligos leads to an overestimation of the diffusion rate of the fast mode, or possibly that our millisecond temporal resolution is not able to detect a much faster short-segment mode of diffusion.

Interestingly, and in contrast to AlkD, AlkF displays a clear bimodal DNA scanning, which deviates significantly from the corresponding monomodal simulated random walk (Fig. 2d, red and black curves). However, any DNA glycosylase activity or base inspection preference has yet to be reported for AlkF[10]; thus, the biological role for this bimodal DNA scanning remains unexplained. The average diffusion rate of the slow and fast modes for AlkF, as classified using the HMM, differs by a factor of 8.25, which corresponds to energy barrier difference of $2.11\,k_BT$. This variation in energy barrier is larger than what was measured for AlkD ($1.46\,k_BT$) and it exceeds the $1.5\,k_BT$ range of helical sliding by 41%, which leads to an explicit bimodality in the overall diffusion rate distribution of AlkF (Fig. 2d, red curve). From the close resemblance with AlkD, the possibility of the formation of an interrogation complex due to base flipping can most likely be ruled out for AlkF as well, hence, demanding further exploration to understand the reason for the observed bimodality.

Based on a model of a possible AlkF–DNA complex[10] (Fig. 2b), we hypothesized that the additional β-hairpin in AlkF, which is not present in AlkD, might protrude into the major groove of the DNA due to its positive charge and play a role in this dual-mode DNA scanning. By mutating three positively charged residues in the β-hairpin and replacing them with neutral residues (Fig. 2c), we observe that the explicit bimodality in the diffusion rate distribution disappears for the loss-of-function mutant AlkF-Δpos (Fig. 2d, red curve) and the diffusion rate distribution resembles both that of AlkD and the simulated random walk (Fig. 2d). Without the charged structural element, AlkF scanning becomes more homogenous and less interrupted with the average diffusion rate increasing more than fourfold, as shown by the shift of the distributions (Fig. 2d, solid black line versus dashed line). Structural superposition of the two molecules in the asymmetric unit for AlkF[10] shows flexibility in this element, with a relative conformational shift of up to 1.1 nm for the position of the charged residues in the β-hairpin (Fig. 2b, inset). This gives enough flexibility to this hairpin to protrude into the DNA groove and retract from it intermittently. In the protruded state, binding of the protein to the DNA is stronger, leading to a higher energy barrier of stepping and lower diffusion rate. In contrast, in the retracted state, the protein might be able to scan along the DNA with a higher diffusion rate and potentially able to hop over the grooves due to decreased electrostatic contact. Such a conformation-regulated DNA-scanning resonates well with the bimodality observed in the diffusion rate distribution. Mutation of the positively charged residues in the β-hairpin (Fig. 2c) lowers its electrostatic potential for collapsing onto the DNA surface, hence altering the observed bimodality of scanning and increasing the diffusion rate for AlkF-Δpos. We conclude that the positively charged residues in the β-hairpin loop play a central role in the dynamic process of DNA scanning, and more specifically the modality of scanning, by AlkF. This rules out the formation of an

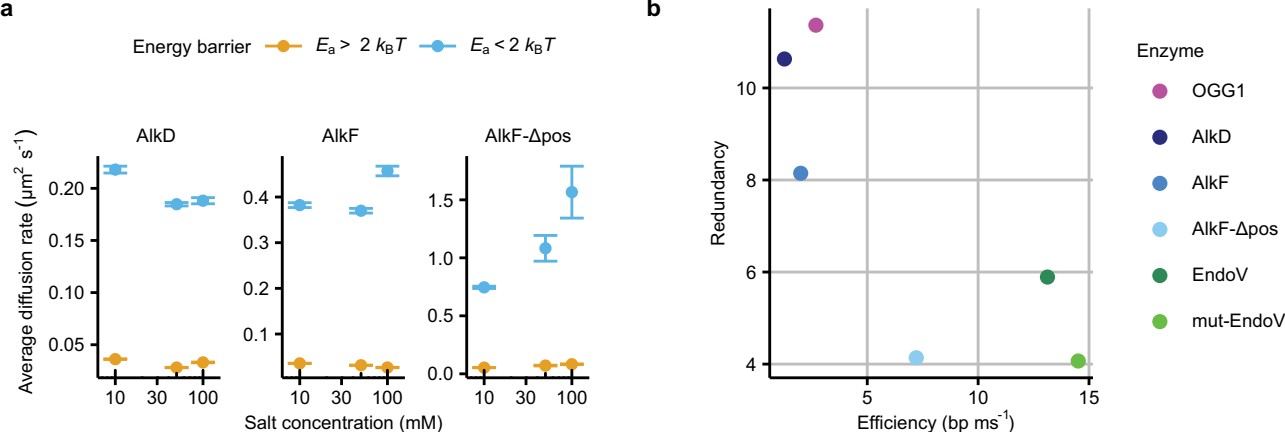

**Fig. 3 Hopping, redundancy, and efficiency of DNA scanning. a** The average diffusion rate for modes of scanning with high (yellow) and low (blue) activation barriers as a function of salt concentration. The error bars show the standard error of the mean (SEM). **b** Comparison of redundancy and efficiency of DNA scanning for six proteins. x axis: efficiency; the average rate at which bases are covered in each binding and scanning event (bp ms$^{-1}$). y axis: redundancy; the average number of times each base pair is visited per binding and scanning event.

interrogation complex and base flipping as potential reasons for the observed bimodality in scanning by AlkF.

The results of the completely independent HMM analysis and the kinetic classification are in good agreement, both with respect to the calculated average diffusion rate and the mode occupancy (Fig. 2e). There is good correspondence between the slow and fast modes of HMM with the high ($E_a > 2 k_B T$) and low ($E_a < 2 k_B T$) activation barrier modes of the kinetic classification, respectively. A confusion matrix of the two classification methods shows 79% accuracy in segmenting the trajectories into the respective modes. Moreover, comparing data collected at different salt concentrations, shows that the removal of the positively charged residues in the AlkF β-hairpin motif results in a salt-dependent trend in the average diffusion rate for the low-energy-barrier mode (Fig. 3a), suggesting that the AlkF-Δpos mutations enable hopping as a fast translocation mode similar to the previous reports[11,18,29]. This salt-dependence is not noticeable for AlkD or wild-type AlkF, hence within the resolution limits of our experiment, hopping does not seem to contribute to efficient coverage of DNA by the HLR proteins.

We have previously investigated the scanning properties of OGG1, wild-type, and wedge-mutant EndoV (EndoV and mut-EndoV)[11]. In order to compare the scanning characteristics for these and the current HLR proteins side-by-side, we calculated the redundancy and efficiency of scanning for all six proteins. Redundancy is defined as the average number of times each base pair is visited in a single binding and scanning event, while the efficiency of scanning is the average rate of scanning in terms of base pairs per millisecond (bp ms$^{-1}$). We calculated these values for the six proteins in question, and the efficiency-redundancy plot (Fig. 3b) reveals that OGG1, AlkD, and AlkF all have relatively low efficiency in covering the length of the DNA (1–3 bp ms$^{-1}$) compared to the other proteins (7–15 bp ms$^{-1}$), yet with approximately twice as high redundancy. We conclude that OGG1, AlkD, and AlkF interact with DNA in confined regions where they check bases with high redundancy as they switch their scanning direction more frequently than it would be expected by random thermally driven diffusion. In contrast to this, the wild-type and wedge-mutant EndoV are four to five times more efficient than OGG1, AlkD, and AlkF, covering 13–15 bp ms$^{-1}$, but at the expense of redundancy, illustrating a trade-off between these two factors for different proteins. This is consistent with the results of a recent study[30] where an intrinsic trade-off is described between the accuracy (redundancy) and speed (efficiency) of

scanning. It was suggested that the weak nature of nonspecific protein–DNA interactions contribute to the ability of the proteins to hop over bases and increase the speed of scanning at the expense of accuracy of target detection. This is exemplified by EndoV, which has the ability to hop as we have shown recently[11], and thus occupies the low redundancy/high-efficiency region of the plot (Fig. 3b). The suggested trade-off is furthermore well demonstrated by comparing EndoV and AlkF with their respective mutants (Fig. 3b). Removal of the wedge in EndoV and the positive patch in the β-hairpin in AlkF increases the efficiency of scanning while substantially decreasing redundancy, implying a role for these elements in accurate base interrogation and/or strong DNA binding.

In a wider perspective, analysis of redundancy and efficiency of DNA scanning might provide a deeper understanding of the roles of different DNA-scanning proteins also beyond functions in repair pathways. For instance, proteins evolved to locate randomly occurring and evenly distributed sites across the genome would be expected to have high efficiency to be able to cover long stretches of DNA without disengaging from the DNA. In contrast, proteins involved in the detection of clustered sites in specific regions would be expected to have high redundancy to be able to scan DNA more accurately within a confined stretch of DNA. Recent observations have shown that the DNA repair protein OGG1 might have roles beyond DNA repair by binding to sites in DNA enriched in 8-oxoG in a non-catalytic way[31–34], or in guanine-rich, potential G-quadruplex-forming sequences[34,35]. These observations are in line with our finding that OGG1 has high redundancy at the expense of efficiency (Fig. 3b). Moreover, detection of bulky lesions/obstacles or lesions causing a larger variation in the energy barrier, such as e.g., weakening of base pairing as observed for substrates relevant for EndoV[36,37], put less demand on redundancy in scanning thus allowing proteins to move faster along DNA.

In conclusion, analysis of trajectories of single-molecules moving along linear DNA shows that the HLR DNA glycosylase AlkD searches for base lesions without forming a stable interrogation complex, in contrast to previously reported multimode scanning by other DNA base repair proteins known to use wedging or intercalating residues in combination with base lesion binding pockets to form stable interrogation complexes. This result resonates well with the unique lack of base flipping for base interrogation, damage recognition, and base removal, which has so far only been observed for AlkD. The structural homolog AlkF, on

the other hand, has an additional positively charged β-hairpin, causing the protein to scan DNA in a dual-mode fashion, through stronger protein–DNA interactions enforced by this positively charged major groove-binding element. Neutralization of this charged structural element in the AlkF-Δpos mutant weakens the nonspecific protein–DNA interaction and makes the energy barrier landscape for scanning smoother, turning this loss-of-function mutant into a faster DNA-scanning protein with the ability to perform hopping. Finally, our data support the argument that an intrinsic trade-off between redundancy and efficiency, facilitated by hopping, exists as a general aspect of DNA scanning.

## Methods

**Preparation of DNA and proteins**. The DNA is anchored to the surface via immobilized streptavidin at one end and to anti-digoxigenin-coated polystyrene beads at the other end. For this purpose, a 12 kbp λ-DNA fragment was produced by PCR with biotin and digoxigenin tags at either end using primers modified with 5′ biotin (5′-bio-ACTTCGCCTTCTTCCCATTT-3′) and 5′ digoxigenin (5′-dig-ATCTCGCTTTCCACTCCAGA-3′) (Eurofins MWG/Operon). In total, 50 µl of the PCR solution contained 300 µM of each dNTP, 0.4 µM of each primer, two units long Amp Taq DNA polymerase (New England Biolabs), and 5 ng of λ-DNA template in a 1× LongAmp Taq Reaction Buffer. The PCR amplification started with an initial denaturation step at 94 °C for 3 min, followed by 35 cycles of denaturation at 94 °C for 15 s, annealing at 60 °C for 60 s and primer extension at 65 °C for 16 min, before ending with a final step with extension at 65 °C for 10 min.

Genomic sequences for full-length *E. faecalis* AlkD and *B. cereus* AlkF were inserted into pET28b in frame with the hexahistidine tag. The AlkF-Δpos mutant, with residues Arg203, Lys206, and Lys207 mutated to alanine, was previously designed by site-specific mutagenesis[10]. The expression vectors pET28b-AlkD, pET28b-AlkF, and pET28b-AlkF-Δpos were transformed into the *E. coli* BL21 CodonPlus(DE3)-RIL strain (Stratagene) for protein expression. Cells were grown in LB-medium supplemented with 50 µg ml$^{-1}$ kanamycin at 37 °C until an $OD_{600}$ of 0.7 was reached, whereupon the expression was induced by the addition of isopropyl-β-D-thiogalactopyranoside (IPTG) to a final concentration of 0.25 mM. Induced cells were grown for an additional 18 h at 18 °C prior to harvesting by centrifugation. Cell pellets were resuspended in lysis buffer containing 50 mM Tris pH 7.5 and 300 mM NaCl, followed by sonication for 3 × 30 s on ice. Insoluble cellular debris was removed by centrifugation at 27,000 × g for 20 min and the supernatant was applied to Ni-NTA resin equilibrated with lysis buffer, and the captured proteins were eluted using a buffer with 300 mM imidazole, 300 mM NaCl, and 50 mM Tris pH 7.5. The purified proteins were dialyzed extensively against 1× PBS buffer prior to labeling with a fluorophore for single-molecule experiments. The proteins were labeled with ATTO-647N (ATTO-TEC) by mixing each protein and fluorescent dye in molar ratio 3:1 in PBS buffer and incubate the mixtures at room temperature for 30 min in the dark. The labeled protein was separated from an unreacted dye by using a buffer exchange/size-exclusion NAP-5 column (GE Healthcare) and 1× PBS buffer. The degree of labeling was determined to be ~50% by absorption spectroscopy using a NanoDrop One instrument (Thermo Scientific). Structure figures were prepared using the PyMol molecular graphics system (Schrödinger Inc.). The surface potential of AlkF and AlkF-Δpos were calculated using the APBS[38] plugin in PyMol, and the contour levels were set to ±3 for the potential at the solvent-accessible surface.

**DNA anchoring and elongation**. Analysis of DNA scanning is performed by observing the movement of proteins along a linear DNA track[11]. Therefore, single DNA molecules were attached to the coverslip surface of the flow chamber by streptavidin–biotin linking. The DNA was then elongated by optically trapping and displacing a polystyrene bead attached to the free end of the DNA via an anti-digoxigenin–digoxigenin link, before finally exposing the DNA to single, fluorescently labeled protein molecules. In order to prepare surface-functionalized coverslips, we built upon adjusted methods from previous reports[11,39–43] and manufacturers' manuals. Untreated coverslips (24 × 60 mm, Menzel Gläser) were placed in glass staining troughs and cleaned three times by alternating between 10 min sonication in 1 M potassium hydroxide and 100% ethanol, interspersed with rinsing in double-distilled (dd) water, followed by a 10 min sonication in 100% acetone. The coverslip surface was then amino-functionalized by incubation in 2% 3-aminopropyltriethoxysilane (Sigma-Aldrich) in acetone for 4 min with a 30 s sonication in the middle of this period. Coverslips were rinsed with dd water, dried with nitrogen gas, and incubated at 100 °C for 30 min, then cooled to room temperature. In all, 120 µl of a solution containing 150 mg ml$^{-1}$ amino-reactive polyethylene glycol (PEG-NHS, MW = 5000 g mol$^{-1}$, Nanocs), 0.1 mg ml$^{-1}$ biotinylated PEG-NHS, and 0.1 M sodium bicarbonate were sandwiched between two amino-functionalized coverslips. The coverslips were kept in a humid chamber overnight. Saturation of the surface with PEG molecules is an essential step for preventing unspecific binding of proteins to the surface, and the inclusion of a small number of biotinylated PEGs provides several binding sites for streptavidin-linked DNA. The PEGylated coverslips were rinsed with dd water, dried with nitrogen gas,

and stored in a vacuum chamber to avoid degradation. The flow cell consists of a microscope slide covered with a double-sided tape with a cut-out channel of 5 × 40 mm connecting two holes cut through the plate. The two holes were connected to a pumping system using PEEK and silicone tubing. Immediately before the final assembly of the flow cell, a 120 µl solution of 0.015 mg ml$^{-1}$ streptavidin (from *Streptomyces avidinii*, Sigma-Aldrich) in PBS buffer was sandwiched between a PEGylated coverslip and a clean coverslip for 5 min, rinsed with dd water and dried in a stream of nitrogen gas. The streptavidin-containing coverslip was placed on the double-sided tape and the flow cell mounted on the microscope. The flow cell was sequentially injected (50 µl min$^{-1}$) with 200 µl of running buffer (25 mM Tris, pH 7.5), 200 µl of blocking buffer (25 mM Tris, pH 7.5, 2 mM EDTA, 1–3 mg ml$^{-1}$ BSA, 0.01% (v/v) Tween-20) and incubated for one hour, followed by 200 µl of 20 ng ml$^{-1}$ DNA in running buffer (10 µl min$^{-1}$) and incubated for 20 min. Then 400 µl of a mix of 10% blocking buffer and 90% running buffer supplemented with 10 µg ml$^{-1}$ beads were added (10 µl min$^{-1}$) and incubated for an hour, allowing the beads to attach to the free ends of immobilized DNA molecules on the coverslip. Unbound beads were washed away with a small volume of running buffer before the optical trap was used to capture a single bead and stretch the DNA to ca 95% of its maximal length. The power of the trapping laser and the bead displacement from the optical trap center was kept constant upon elongation of the DNA, ensuring that the exerted force on the DNA remained constant throughout all experiments. Finally, fluorescently labeled protein molecules were added, the flow stopped, and signals from DNA-scanning molecules were recorded.

**Optical setup**. The optical setup used in this study has been described previously[13]. Briefly, two important capabilities of the setup are optical trapping and high spatial- and temporal-resolution fluorescence single-molecule detection. The holographic optical tweezers were built on an inverted microscope (IX-71; Olympus) by coupling an infrared laser (MIL-H-1064; CNI) with the wavelength of 1064 nm onto the back focal plane of a microscope objective lens (PlanApo, X60, NA 1.45; Olympus). A spatial light modulator (XY Series 512 × 512; Boulder Nonlinear Systems) was used to steer the optical trap. Fluorescence excitation was achieved using 488- and 647-nm laser light emitted by an argon–krypton ion laser (Innova 70 C; Coherent). In order to avoid noise from the full volume of the sample, the excitation was performed in highly inclined and laminated optical sheet mode[44] (HILO) by adjusting the position of the excitation beam entering the objective lens. Fluorescence emission was collected by the same objective lens, magnified by and projected onto an EMCCD camera (iXon DV887DCS-BV; Andor). The data were collected as sequences of images (frames) with a frequency of 30–75 Hz using Micro-Manager software[45]; this resulted in 120–200 photons detected per single-molecule per frame on average.

**Data collection, image processing**. After exposing the DNA to fluorescently labeled proteins, we observe the proteins scanning the DNA and record the trajectories of movement in real time. Supplementary Movies 1–3 show concatenation of exemplary protein trajectories along DNA visualized using the TrackMate plugin[46]. All signals in each frame originating from both free-floating proteins, proteins stuck to the surface, and proteins scanning the DNA were localized by Gaussian fitting using the ThunderSTORM[47] plugin in FIJI[48]. Thereafter, using custom-written code[49] in R based upon previous reports[11,50], the trajectories of molecules moving along the DNA were detected among all trajectories. The short-lived trajectories from free proteins, moving in and out of the illumination field in the z direction, were excluded by filtering out signals lasting fewer than five consecutive frames. Due to the nature of Brownian motion, free proteins also move completely randomly in the x–y plane, in contrast to DNA-scanning proteins, which follow a straight line radiating from the bead in the optical trap. Instances of DNA-scanning proteins were detected by setting a filter to include the signals from molecules moving at least 600 nm (over 100 times their diameter) along a linear path. Using the trace of ten of these trajectories for each continuously recorded data set, the position of the unlabeled DNA was determined, and a rotation matrix was applied to align the DNA along the x axis. As the final step, signals from DNA-scanning proteins were separated from signals from free-floating and surface-bound proteins by selecting only signals lasting longer than five consecutive frames, and moving at least 300 nm in the x direction, and at the same time being within 200 nm of the calculated DNA position in the y direction. The number of detected trajectories for AlkD, AlkF, and AlkF-Δpos adds up to 1049, 659, and 470, respectively. The spatial precision, determined by looking at stationary proteins bound directly to the coverslip surface, was calculated to be 18.7 ± 3.0 nm, which leads to an error of 0.0066 ± 0.0060 µm² s$^{-1}$ in the calculation of the diffusion rate.

**Instantaneous diffusion analysis**. Based on an initial visualization of the raw trajectory data (Fig. 1, lower panel and Supplementary Movies 1–3), an apparent heterogeneity can be observed that may reflect the different conformations or modes by which proteins scan the DNA. To examine the existence of meaningful modalities within this apparent heterogeneity, the instantaneous diffusion rate in all frames over all protein trajectories was calculated in order to analyze the distribution. For this purpose, a moving window of five frames was used, and for each frame, the mean squared displacement (MSD) ($<x^2>$) of the next five consecutive frames was calculated. Using the diffusion coefficient equation for one-dimensional

diffusion, $D = <x^2>/2t$ where $t$ is time, the diffusion coefficient ($D$) was calculated and assigned to that particular frame as an instantaneous diffusion rate. This process was repeated for all frames of all trajectories while moving the window through all trajectories. The distribution of these calculated values for the instantaneous diffusion rate for the different proteins as well as that of their respective random walks is referred to as diffusion rate distribution. By changing the width of the window to 7, 10, and 15 frames, we verified that the main features of the distribution of the instantaneous diffusion rates are independent of the width of the window. Since the instantaneous diffusion rate is calculated for each frame, the average diffusion rate of any segment can be calculated simply by averaging the instantaneous diffusion rates for all the frames within that particular segment. In addition to the real protein trajectories, three single-mode random walks with average diffusion rates equal to each of the three proteins were simulated. No effect of protein–DNA interaction is implemented in these simulations, therefore the instantaneous diffusion rate plots represent single-mode distributions, showing how the scanning of DNA by these proteins would have looked like under the assumption of a homogenous energy landscape when binding to the DNA. Therefore, any deviation from this pure random diffusion is attributable to the nature of protein–DNA interactions.

**Classification of trajectories based on kinetics of scanning**. The energy barrier method looks at the kinetics of molecule movements to estimate the local energy barrier that limits free diffusion. This energy barrier for proteins to step from one base to the next is highly dependent upon the relative affinity between the protein and DNA for that position which might be affected by conformational variations in the protein or DNA or in the protein–DNA contacts. The high affinity of protein active sites to inspect specific bases creates a high-energy barrier for proteins to move away from that base. In contrast, low-affinity creates a smoother landscape for proteins to scan along the DNA. Hence, variations in this energy barrier are indicative of a modality of scanning that can be regulated by conformational variations. Interaction of proteins with the DNA can be modeled as a kinetic reaction with a rate constant equal to the rate at which the protein moves from one base pair to the next. Therefore, the rate constant of this reaction can be calculated as $k = \frac{1}{t} = 2D/<x^2>$, where $D$ is the diffusion constant and $<x^2>$ is the MSD of the protein along the DNA. Using the calculated rate constant combined with the Arrhenius equation, the activation barrier for scanning from one base to the next is given by $E_a = \ln(\frac{D_{ideal}}{D})k_B T$, where $D$ is the diffusion constant for helical sliding and $D_{ideal}$ is the theoretical value for the diffusion constant of sliding when the activation barrier is zero. From the theoretical upper limits of the diffusion rates for the simulated random walks (0.49 and 0.80 $\mu m^2\,s^{-1}$ for AlkD and AlkF/AlkF-$\Delta$pos, respectively), and the experimental values for the instantaneous diffusion rates, the activation barrier $E_a$ was calculated for the collected protein trajectories. It has been suggested that proteins scan DNA either in search, recognition, or intermediate interrogation mode, each mode having distinct ranges of activation barriers $E_a$ for diffusive movement[20,27]. In the search mode, proteins mainly follow the DNA helix in a smooth binding-energy landscape with roughness below $2\,k_B T$[11,14–21], however, interspersed periods of hopping in this mode have been reported[11,16,18,29,51–55], allowing for fast and efficient coverage of the DNA. In the pre-recognition (or interrogation) mode[11,25,27], the binding-energy landscape is rougher ($E_a > 2\,k_B T$)[20], and movement of the proteins along DNA is considerably slower[11,21,56,57], allowing for close inspection of individual bases before transitioning to recognition mode by stably binding to the target base ($E_a > 5\,k_B T$)[20]. Based on the calculated values of the energy barrier for each step of the trajectories and we segmented the trajectories for movements with $E_a < 2\,k_B T$ and $E_a > 2\,k_B T$ that are representative of freely scanning and interrupted movement, respectively.

**Classification of trajectories based on a hidden Markov model (HMM)**. To investigate the observed scanning bimodality from the diffusion rate distribution and the kinetic classification, with a purely statistical method, we compared the scanning of these proteins using hidden Markov model[22] as an unbiased model for classification of the trajectories. As implemented in the single-particle-tracking software vbSPT[22], the HMM model relies on variational Bayesian treatment of a hidden Markov model where it is assumed that the protein performs memoryless transitions between different modes of scanning. The number of modes of movement was specified to be two, classifying each frame of all trajectories into either a slow- or fast-moving mode. Time series of proteins position along DNA are segmented into either fast or slow mode, and using a simple diffusion model, the diffusion coefficients are calculated for each segment. The calculated value of diffusion coefficient for different segments are used as observable components of the Markov chain. The initial values of average diffusion rates and occupancies of each state along with their transition probabilities are calculated as parameters of the model according to the initial segmentation. Using a maximum-evidence criterion, parameters of the model are estimated for the best fit to the experimental data and segmentation of trajectories are finalized. These segmentations are performed independently for each protein, hence the slow and fast modes of each protein are attributed to the fraction of movement with relatively lower and higher diffusion rate for that protein, respectively. Therefore, slow and fast modes of one protein do not necessarily correlate with those of other proteins. The difference between the energy barrier of the two populations classified by HMM analysis for different proteins was calculated using $\Delta E_a = \ln(\frac{D_{fast}}{D_{slow}})k_B T$, where $D_{fast}$ and $D_{slow}$

represent the average diffusion rate of the fast and slow modes of the HMM analysis respectively. Moreover, we compared the overlap between the two segmentation models with a confusion matrix in which slow and fast modes of HMM analysis correspond to high and low-energy barriers of the kinetic analysis, respectively. This analysis shows 79% accuracy in segmentation overlap between the corresponding modes of the two models with sensitivity and specificity of 86% and 73%, respectively.

**Reporting summary**. Further information on research design is available in the Nature Research Reporting Summary linked to this article.

## Data availability

All data including the raw image data captured for this study are available from the corresponding authors upon reasonable request.

## Code availability

All source codes used in the analysis pipeline are available on GitHub: https://doi.org/10.5281/zenodo.5001380.

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

## Acknowledgements

We acknowledge South-East Norway Regional Health Authorities (grants 2014034 and 2015095 to B.D.), the Research Council of Norway (program FRIMEDBIO to M.B., program DAAD project 281255 to B.D.), the German Academic Exchange Service (grant no. 57402206 to M.S.), and the University of Oslo, MLS^UiO program and Medical Faculty (grants to A.A.) for funding.

## Author contributions

A.A., M.S., M.B., A.D.R., and B.D. designed the study; A.A., P.H.B., P.B., and B.D. prepared DNA and proteins; A.A. and K.T. performed single-molecule experiments, with contributions from R.D. and M.S.; A.A., K.T., K.G., J.T., M.B, B.D., and A.D.R. participated in the flow-cell design; A.A. analyzed the data, with contribution from A.D.R.; A.A. and B.D. interpreted and presented the results, with contributions from P.H.B., M.S., M.B., and A.D.R.; A.A. and B.D. wrote the paper with contributions from all the other authors.

## Competing interests

The authors declare no competing interests.
