## [Peer Review File · Communications Biology]

Reviewers' Comments:

Reviewer #1:

Remarks to the Author:

This manuscript utilizes single molecule fluorescence microscopy methods to show that a bacterial glycosylase, AlkD, diffuses along DNA with only a fast diffusion mode. The authors also characterize the diffusive behavior of an AlkD structural homolog, AlkF, and identify structural features of AlkF that are responsible for a slow diffusive mode. The authors utilize hidden Markov modeling and an energy barrier-limited kinetic model to extract further mechanistic information about diffusion of AlkD and AlkF.

This paper used elegant methods to characterize an atypical search mechanism for a DNA repair enzyme. The methods follow closely to those published recently by the authors (Ahmadi, et al., Nature Communications 2018) to study EndoV and hOGG1. The authors use nice figures to illustrate the data, and for the most part, the data support the conclusions in the paper. I especially liked the efficiency/redundancy analysis. However, I had some questions about the manuscript that may be addressed with further revisions.

1. I found it difficult to follow the methods used in the manuscript until I read the previous paper published by the same authors (Ahmadi, et al., Nature Communications, 2018). For this reason, I was wondered whether this paper can currently stand alone in a biology journal. Even though this is in a short communication format, the paper could use little bit more detail describing the instantaneous diffusion methods, the hidden Markov method, and the energy barrier method and why each is used.

2. Another single molecule fluorescence paper was published recently on AlkD (Peng, et al., PNAS, 2020) that suggested two AlkD diffusion modes. I think that the results in the current manuscript should be contextualized with the conclusions from this paper from Peng, et al. The single molecule assay in Peng, et al. is significantly different from the one used here, and it is likely that the use of longer DNA in the current work provides different insight into the AlkD search mechanism. I think this should be discussed in the manuscript.

3. The title of the paper states that AlkD uses a single mode of motion, yet the HMM analysis in Figure 1d shows two distinct populations. How are the authors defining a diffusion "mode?" Does a mode have to show a distinct peak in the instantaneous diffusion curve, or is it possible that AlkD has two modes with slight differences in diffusion constant that are unresolved in the instantaneous diffusion curve?

4. At first, I found it difficult to understand the relevance of AlkF to this work. My conclusion was that AlkF was an AlkD structural homolog, which leads me to wonder why DNA scanning behavior should correlate between the two proteins. The rationale for comparing AlkF diffusion to AlkD diffusion should be clearly described in the introduction of the paper. Is it likely that glycosylase activity will be found for AlkF? Is there any other putative function for AlkF? Also, why didn't the authors clone the beta-hairpin out of AlkF to make it into AlkD?

5. In line 93-94, the authors state "we conclude that the positively charged residues in the beta-hairpin motif contribute to a strong interaction between the protein and DNA, which affects the dynamic process of DNA scanning." I am curious whether the authors could speculate how, mechanistically, interactions of this positive loop with DNA might contribute to the slow diffusion AlkF. Also, how might this relate to function of AlkF?

6. The efficiency/redundancy analysis is fun and interesting, and contextualizes the current results with the previous work by Ahmadi, et al. It would be interesting for a broad audience to expand this section to talk about what might be the functional reason for some of these enzymes to have high redundancy or high efficiency. For example, does it relate to the type and frequency of the specific

lesion? Does it relate to copy number?

Reviewer #2:

Remarks to the Author:

I was unable to review this manuscript because figures presenting the raw data and the processed autocorrelation plots were not included. In my opinion, this manuscript should be sent back to the authors, to add those items prior to review.

Reviewer #3:

Remarks to the Author:

Ahmadi et al. present a single-molecule analysis of DNA scanning by the HEAT-like repeat (HLR) AlkD glycosylase and the structurally related HLR protein AlkF. AlkD has been shown to be a DNA glycosylase that does not use "base flipping" to survey the genome in search of a lesion, likely because it has evolved to remove bulky lesions such as N3-yatakemycinyladenine, which would not fit into an active site. AlkF, on the other hand, is less well characterized but has been shown by this same group to have the same overall structure as AlkD but to bind preferentially to 3- and 4-way (Holliday) junctions. Here, the authors report that AlkD uses a single mode of scanning, whereas AlkF uses a dual mode of fast and slow scanning modes. The authors show that by removing a beta hairpin unique to AlkF, the scanning mode becomes monomodal like AlkD, collapsing to only the fast mode. The authors attribute this to the fact that the positively charged residues on the beta hairpin slow the protein down by interacting with the DNA backbone, and speculate that removing this motif enables a faster scanning mode of hopping.

The question of how DNA repair proteins scan DNA and recognize damage is an important and largely poorly understood question. Moreover, this particular class of DNA glycosylase is unlike any other and is a worthwhile system to study in its own right and in comparison to other glycosylase families. Thus, this work should be of interest to a relatively broad audience. The data are of high quality and the manuscript is relatively well written. However, I have a few issues that should be resolved prior to publication.

I am not completely convinced in the interpretation that AlkD scans by a single mode. Looking at the histograms of slow and fast trajectories in Fig 1d, there are clearly two modes for both AlkD and AlkF, despite the overall density distribution reflected in the red curve. I also do not see a significant difference between AlkD and AlkF plots in Fig. 2e. So, there seems to be a fine line between single- and dual-mode definitions used here. Related, I am not sure I understand the statement, "However, classification of the trajectories into fast and slow modes using Markov model analysis indicates the existence of an intermittent state in DNA scanning, which is consistent with a recent simulation study(16) where it is suggested that subtle intermediate states could exist due to conformational shifts observed in the AlkD-DNA contact" (lines 64-67). What is the feature that indicates the intermittent state here?

While there is a very subtle difference between AlkD and AlkF data in Fig. 2, the perturbation in fast and slow trajectories of AlkF by the b-hairpin mutant is very striking. However, the this group previously showed that AlkF has a very weak affinity for duplex DNA and instead is specific for 3- and 4-way junctions (Backe et al, J Struct Biol (2013) 183: 66-75). How to the authors reconcile the two seemingly contradictory studies? While the AlkF result is much clearer, it is maybe not as impactful as the AlkD result given our relative understanding of each protein.

Minor points:

1. The title is grammatically incorrect as written, unless you assume "AlkD" is plural. I believe "scan"

should be "scans". The same is true in the abstract on line 18 (should be "...AlkD detects and excises bases...") and on line 136 (should be "...AlkD searches..")

2. Given the relatively small number of papers that have been published on AlkD, the citations in the first and second paragraphs need to be improved.

a. Lines 27-29: "Four of these are present in human cells, while the HEAT-like repeat family has only been detected in bacteria, archaea and some simple uni- and multicellular eukaryotes (3)." Three additional studies performed bona fide phylogenetic analyses and should be cited: Alseth et al, *Mol Microbiol* 2006 (current reference #6), Rubinson et al, *Nature* 2010 (current reference #4), and Shi et al, *BioEssays* (2018) 40, 1800133.

b. Lines 35-37: "AlkD removes the lesions N3-yatakemycinyldeoxyadenosine (d3yA), N3-methyl-2'-deoxyadenosine (d3mA) and N7-methyl-2'-deoxyguanosine (d7mG) from DNA,..." In addition to references #5 and #6, the authors may wish to add Shi et al, *BioEssays* (2018) 40, 1800133, which reviews the substrates removed by the HLR glycosylases.

c. Lines 37-38: "...and in the case of the bulky d3yA adduct, damage recognition takes place by interaction with the phosphoribose backbone" is not entirely correct. The crystal structure of AlkD in complex with d3yA DNA (Mullins et al, *Nat Chem Biol* (2017) 13: 1002-1008) showed that d3yA was recognized by AlkD interactions with both the d3yA and the DNA backbone. The sentence should therefore be changed to "...and in the case of the bulky d3yA adduct, damage recognition takes place by interaction with both the phosphoribose backbone and the d3yA compound."

d. Lines 38-40: "The smaller and intrinsically labile d3mA and d7mG bases are conversely believed to be removed by hydrolysis due to increased lifetime of these bases in a solvent-exposed orientation (4)." This hypothesis was disproven by later papers from the same lab, and thus this sentence should be replaced with, "The smaller and intrinsically labile d3mA and d7mG bases are conversely believed to be removed by hydrolysis due to stabilization of the increased positive charge on the deoxyribose backbone through electrostatic and CH-pi interactions (Mullins et al, *Nature* (2015) and Parsons et al, *JACS* (2016) 138, 11485–11488)."

3. Line 36: N3-yatakemycinyldeoxyadenosine (d3yA) should be N3-yatakemycinyl-2'-deoxyadenosine (d3yA) to be consistent with the d3mA and d7mG nomenclature used.

4. Line 161, why was *E. faecalis* AlkD used rather than *B. cereus* AlkD, which has been used in most previous studies?

We appreciate the critical review and the perspective provided by reviewer #1 and #3 which we found valuable and important in deepening the understanding and improving the presentation of our data. Before presenting the point-by-point response we would like to reflect on one common issue that was raised by both reviewers.

The term “single-mode scanning” that we used to describe AlkD, was mainly an attempt to contrast the scanning of this protein with what we had previously observed for EndoV and hOGG1, where two explicit peaks in the overall instantaneous diffusion distributions were observed prior to application of HMM analysis. There, we concluded that the peak in the lower range of the diffusion rate is due to formation of an interrogation complex that includes flipping of the bases into base recognition pockets and/or probing the bases using a wedge motif. Absence of that additional peak for AlkD led us to interpret the movement of AlkD as a relatively smooth single-mode sliding in which no interrogation complex is stably formed. Although the diffusion rate profile of AlkD resembles a single-mode random walk, we agree with reviewers that using the term “single-mode” to describe the scanning of AlkD is in contradiction with the results of the HMM analysis. Therefore, we have addressed this issue by focusing on the *absence of an interrogation complex* rather than referring to the scanning of AlkD as being single-mode. Moreover, we realized that a comparison of modes detected by HMM for AlkD with those of other proteins (by looking at energy barrier differences between the modes), gives a more quantitatively and functional understanding of the contrast between the scanning of these proteins. This reconsideration led us to change the title of the paper and rewrite some of the interpretations that were addressed by the reviewers. Details are given below as point-by-point response to reviewers’ remarks.

Reviewer #1 (Remarks to the Author):

This manuscript utilizes single molecule fluorescence microscopy methods to show that a bacterial glycosylase, AlkD, diffuses along DNA with only a fast diffusion mode. The authors also characterize the diffusive behavior of an AlkD structural homolog, AlkF, and identify structural features of AlkF that are responsible for a slow diffusive mode. The authors utilize hidden Markov modeling and an energy barrier-limited kinetic model to extract further mechanistic information about diffusion of AlkD and AlkF.

This paper used elegant methods to characterize an atypical search mechanism for a DNA repair enzyme. The methods follow closely to those published recently by the authors (Ahmadi, et al., Nature Communications 2018) to study EndoV and hOGG1. The authors use nice figures to illustrate the data, and for the most part, the data support the conclusions in the paper. I especially liked the efficiency/redundancy analysis. However, I had some questions about the manuscript that may be addressed with further revisions.

1. I found it difficult to follow the methods used in the manuscript until I read the previous paper published by the same authors (Ahmadi, et al., Nature Communications, 2018). For this reason, I was wondered whether this paper can currently stand alone in a biology journal. Even though this is in a short communication format, the paper could use little bit more detail describing the instantaneous diffusion methods, the hidden Markov method, and the energy barrier method and why each is used.

The original manuscript was written in a very condensed format. We have now expanded the manuscript with more details on the methods by

- Including another figure to the manuscript (Fig. 1) depicting the setup and an exemplary trajectory of AlkD scanning.
- Providing 3 supplementary videos including exemplary trajectories for each protein.
- Incorporating key remarks about the experimental setup, the instantaneous diffusion method, the energy barrier method, and the hidden Markov method (lines 69-101).
- Expanding and rewriting the methods section with separate headlines dedicated to each topic mentioned by the reviewer (352-423).

In addition, we can make all the analysis code and data available for reviewers, and they can be made publically available upon publication.

2. Another single molecule fluorescence paper was published recently on AlkD (Peng, et al., PNAS, 2020) that suggested two AlkD diffusion modes. I think that the results in the current manuscript should be contextualized with the conclusions from this paper from Peng, et al. The single molecule assay in Peng, et al. is significantly different from the one used here, and it is likely that the use of longer DNA in the current work provides different insight into the AlkD search mechanism. I think this should be discussed in the manuscript.

We appreciate the reviewer for bringing this paper (Peng et al., 2020) to our attention. The fact that they have observed a form of bimodality, also in line with predictions in another recent simulation study on AlkD (Votaw & McCullagh, 2019) is consistent with our HMM analysis showing that AlkD has different modes for scanning DNA despite not forming a stable interrogation complex.

In a more in-depth comparison of our results with the Peng et al paper, we looked into the range of diffusion rates reported for the two different modes. The average diffusion rate for the slow mode of AlkD in our experiments is $0.043 \mu\text{m}^2\text{s}^{-1}$ which is within the $0.007 - 0.058 \mu\text{m}^2\text{s}^{-1}$ range that was reported by Peng et al. However, the diffusion rate estimated for the fast mode of AlkD in the Peng study is $0.92 \mu\text{m}^2\text{s}^{-1}$, which is around 5-fold higher than our rate at $0.18 \mu\text{m}^2\text{s}^{-1}$. Given the substantial differences between our two experimental approaches, two plausible explanations can possibly explain the discrepancy: (1) Difference in DNA substrate length; (2) difference in temporal resolution. We have included a short paragraph comparing and discussing the relevant points between these three studies (lines 124-144).

3. The title of the paper states that AlkD uses a single mode of motion, yet the HMM analysis in Figure 1d shows two distinct populations. How are the authors defining a diffusion “mode?” Does a mode have to show a distinct peak in the instantaneous diffusion curve, or is it possible that AlkD has two modes with slight differences in diffusion constant that are unresolved in the instantaneous diffusion curve?

Based on the HMM analysis, AlkD has two modes with subtle differences in the diffusion constant that cannot be resolved in the overall instantaneous diffusion rate distribution. As mentioned in the introductory remark, although the diffusion rate distribution of AlkD resembles that of a single mode random walk, we should not refer to scanning of AlkD as being single mode as pointed out. We have changed the wording both in the title, abstract and main text to reflect this.

The formation of stable interrogation complexes by OGG1 and EndoV in our previous work lead to explicit bimodal overall diffusion distributions, with an energy barrier difference of $2.30 k_B T$ between the two modes. The absence of a peak in the lower range of the diffusion rate for AlkD indicates absence of a corresponding interrogation complex. Indeed, the difference between the energy barriers of the two resolved modes of the HMM analysis is only $1.46 k_B T$, which is considerably lower than the energy barrier difference of the two modes hOGG1 and EndoV. We

conclude that although two modes are resolved for AlkD by HMM analysis, they are not as distinct as the ones formed by stable interrogation complexes, and we refer to them as intermediate states. We have addressed this point in lines 102-123.

4. At first, I found it difficult to understand the relevance of AlkF to this work. My conclusion was that AlkF was an AlkD structural homolog, which leads me to wonder why DNA scanning behavior should correlate between the two proteins. The rationale for comparing AlkF diffusion to AlkD diffusion should be clearly described in the introduction of the paper. Is it likely that glycosylase activity will be found for AlkF? Is there any other putative function for AlkF? Also, why didn't the authors clone the beta-hairpin out of AlkF to make it into AlkD?

Both AlkD and AlkF belong to the HLR DNA binding proteins, although only AlkD has been assigned a catalytic function. Since they both contain the HLR motif, bind DNA, and lack a typical base-binding pocket, we decided to include AlkF in this study to understand the effect of non-base-flipping interrogation on scanning, and to establish a correlation between protein fold, lack of flipping and DNA scanning. As it turned out, the AlkF behaved quite different due to the presence of the additional beta hairpin. We have included an explanation in the introduction as to why we wanted to compare AlkF diffusion with AlkD (lines 49-56).

There is no other putative function assigned to AlkF as far as we know. Despite testing a series of substrates, including alkylated bases (3mA, 7mG, ethenoadenine), oxidized bases (faPy, 8oxoG, 5-oh-C), and deaminated bases (uracil, inosine) no glycosylase activity has been detected. It cannot be excluded that AlkF is highly specialized to remove one or more rare modifications in DNA – similar to AlkD – however, even AlkD has a wider spectrum and was first detected by removal of the more common modifications 7mG, 3mG and 3mA.

The beta-hairpin in AlkF is linking two alpha-helices in the HLR motif, and the geometries in AlkF and AlkD are slightly different. Replacing the long hairpin with a shorter loop, or a long flexible glycine-rich loop, was not done since we worried this could compromise the 3D-folding and relative positions of alpha-helices in the HLR core of AlkF, rendering the scanning data less valuable. We thus decided to only remove charges at the tip of the hairpin to verify our hypothesis on the effect of these positively charged residues with respect to scanning bimodality.

5. In line 93-94, the authors state “we conclude that the positively charged residues in the beta-hairpin motif contribute to a strong interaction between the protein and DNA, which affects the dynamic process of DNA scanning.” I am curious whether the authors could speculate how, mechanistically, interactions of this positive loop with DNA might contribute to the slow diffusion AlkF. Also, how might this relate to function of AlkF?

Inspection of the beta-hairpin in the crystal structure of AlkF reveals two different conformations of the motif (there are two crystallographically independent molecules in the crystal). We have included a figure (Fig. 2b inset) showing the flexibility of the hairpin, which might provide an explanation for the two different modes of scanning. When the hairpin is close to the negatively charged DNA surface, the energy barrier for stepping might be higher and diffusion rate lower than when the hairpin is in the second conformation. Of course, these low-energy conformations might not be the exact conformations found in the AlkF-DNA complex as the crystal structure does not contain DNA, but at least the existence of flexibility in the beta-hairpin offers a putative mechanistic explanation for the observed bimodality. We have addressed this point in lines 176-187.

6. The efficiency/redundancy analysis is fun and interesting, and contextualizes the current results with the previous work by Ahmadi, et al. It would be interesting for a broad audience to expand this

section to talk about what might be the functional reason for some of these enzymes to have high redundancy or high efficiency. For example, does it relate to the type and frequency of the specific lesion? Does it relate to copy number?

We appreciate the reviewer's positive feedback on this topic. We found such discussion interesting as well and have expanded that part of the discussion (lines 222-234). We will particularly highlight the hypothesis that OGG1, with low efficiency and high redundancy, matches with a role of OGG1 beyond DNA repair, where the enzyme has been involved in gene regulation by binding to 8-oxoG in regulatory regions rich in guanine with potential enrichment of 8-oxoG - thus a higher redundancy/precision might be a critical factor for OGG1.

Reviewer #2 (Remarks to the Author):

I was unable to review this manuscript because figures presenting the raw data and the processed autocorrelation plots were not included. In my opinion, this manuscript should be sent back to the authors, to add those items prior to review.

We have included an additional figure explaining the setup and an exemplary trajectory (kymograph) of the proteins, in addition to three supplementary videos showing exemplary raw data for each protein. Moreover, we can make all the analysis code with the raw data available to reviewers. We have not used any autocorrelation calculations in this study.

Reviewer #3 (Remarks to the Author):

Ahmadi et al. present a single-molecule analysis of DNA scanning by the HEAT-like repeat (HLR) AlkD glycosylase and the structurally related HLR protein AlkF. AlkD has been shown to be a DNA glycosylase that does not use "base flipping" to survey the genome in search of a lesion, likely because it has evolved to remove bulky lesions such as N3-yatakemycinyladenine, which would not fit into an active site. AlkF, on the other hand, is less well characterized but has been shown by this same group to have the same overall structure as AlkD but to bind preferentially to 3- and 4-way (Holliday) junctions. Here, the authors report that AlkD uses a single mode of scanning, whereas AlkF uses a dual mode of fast and slow scanning modes. The authors show that by removing a beta hairpin unique to AlkF, the scanning mode becomes monomodal like AlkD, collapsing to only the fast mode. The authors attribute this to the fact that the positively charged residues on the beta hairpin slow the protein down by interacting with the DNA backbone, and speculate that removing this motif enables a faster scanning mode of hopping.

The question of how DNA repair proteins scan DNA and recognize damage is an important and largely poorly understood question. Moreover, this particular class of DNA glycosylase is unlike any other and is a worthwhile system to study in its own right and in comparison to other glycosylase families. Thus, this work should be of interest to a relatively broad audience. The data are of high quality and the manuscript is relatively well written. However, I have a few issues that should be resolved prior to publication.

I am not completely convinced in the interpretation that AlkD scans by a single mode. Looking at the histograms of slow and fast trajectories in Fig 1d, there are clearly two modes for both AlkD and AlkF, despite the overall density distribution reflected in the red curve. I also do not see a significant difference between AlkD and AlkF plots in Fig. 2e. So, there seems to be a fine line between single-

and dual-mode definitions used here. Related, I am not sure I understand the statement, “However, classification of the trajectories into fast and slow modes using Markov model analysis indicates the existence of an intermittent state in DNA scanning, which is consistent with a recent simulation study(16) where it is suggested that subtle intermediate states could exist due to conformational shifts observed in the AlkD-DNA contact” (lines 64-67). What is the feature that indicates the intermittent state here?

As explained in the introductory remark of this letter, and also to the Remark #3 by reviewer #1, we appreciate the critical point raised by the reviewer about referring to scanning of AlkD as single-mode and have addressed it thoroughly in the manuscript and particularly in lines 102-123.

Instead of using single-mode as a definitive term to describe the scanning of AlkD, we focused more on the contrast between the energy barrier differences of the observed modes. The approximate $2.30 k_B T$ energy barrier difference between the two peaks for EndoV and hOGG1 that leads to a bimodality in the overall instantaneous diffusion rate distribution, indicates formation of a relatively stable interrogation complex, in line with crystal structures showing flipping of bases. In contrast, an energy barrier difference of only $1.46 k_B T$ between the two modes of the HMM analysis for AlkD, combined with lack of a bimodality in the overall distribution, indicates absence of an interrogation complex for AlkD.

Our revised interpretation above is consistent with the conclusions of the mentioned theoretical study (ref 16; now ref 27) where it is suggested that unlike glycosylases capable of base flipping, AlkD does not form a stable interrogation complex. However, small conformational changes in the AlkD-DNA contact in the search mode observed in the same study suggests existence of subtle intermediate states that are not as distinct as the interrogation states formed by other glycosylases, and this is consistent with the detection of two modes by HMM analysis with lower energy barrier difference for AlkD. We have tried to clarify the quoted sentence by reviewer in lines 124-130.

The energy barrier difference between the two modes of AlkF is around $2.11 k_B T$ which leads to an explicit bimodality in the instantaneous diffusion rate distribution. This shows that the scanning is not as smooth as that for AlkD with mode barrier difference of $1.46 k_B T$ and no explicit bimodality in the diffusion rate profile. This has been addressed in lines 162-165.

While there is a very subtle difference between AlkD and AlkF data in Fig. 2, the perturbation in fast and slow trajectories of AlkF by the b-hairpin mutant is very striking. However, the this group previously showed that AlkF has a very weak affinity for duplex DNA and instead is specific for 3- and 4-way junctions (Backe et al, J Struct Biol (2013) 183: 66-75). How to the authors reconcile the two seemingly contradictory studies? While the AlkF result is much clearer, it is maybe not as impactful as the AlkD result given our relative understanding of each protein.

The function of AlkF is not known, and it is also unclear if the affinity for branched DNA substrates as presented in our 2013 paper does reflect a biological function, or if it is merely serendipity that AlkF ‘gets stuck’ at branching points in DNA. Despite testing a series of substrates including alkylated, oxidized and deaminated bases, AlkF has not shown any glycosylase activity. One could speculate that this protein is highly specialized to remove one or more rare modifications in DNA or has functions outside of DNA base repair, e.g. in recombination, or even beyond DNA repair. However, disclosing the biological role of AlkF is beyond the scope of the present study, and unfortunately, our single-molecule data does not give clues to the biological role of AlkF.

Minor points:

1. The title is grammatically incorrect as written, unless you assume “AlkD” is plural. I believe “scan” should be “scans”. The same is true in the abstract on line 18 (should be “...AlkD detects and excises bases...”) and on line 136 (should be “...AlkD searches..”)

The point has been addressed.

2. Given the relatively small number of papers that have been published on AlkD, the citations in the first and second paragraphs need to be improved.

We have updated the introduction with respect to the references

a. Lines 27-29: “Four of these are present in human cells, while the HEAT-like repeat family has only been detected in bacteria, archaea and some simple uni- and multicellular eukaryotes (3).” Three additional studies performed bona fide phylogenetic analyses and should be cited: Alseth et al, Mol Microbiol 2006 (current reference #6), Rubinson et al, Nature 2010 (current reference #4), and Shi et al, BioEssays (2018) 40, 1800133.

Addressed in line 37.

b. Lines 35-37: “AlkD removes the lesions N3-yatakemycinyldeoxyadenosine (d3yA), N3-methyl-2'-deoxyadenosine (d3mA) and N7-methyl-2'-deoxyguanosine (d7mG) from DNA,...” In addition to references #5 and #6, the authors may wish to add Shi et al, BioEssays (2018) 40, 1800133, which reviews the substrates removed by the HLR glycosylases.

Addressed in line 45.

c. Lines 37-38: “...and in the case of the bulky d3yA adduct, damage recognition takes place by interaction with the phosphoribose backbone” is not entirely correct. The crystal structure of AlkD in complex with d3yA DNA (Mullins et al, Nat Chem Biol (2017) 13: 1002-1008) showed that d3yA was recognized by AlkD interactions with both the d3yA and the DNA backbone. The sentence should therefore be changed to “...and in the case of the bulky d3yA adduct, damage recognition takes place by interaction with both the phosphoribose backbone and the d3yA compound.”

Addressed in line 46.

d. Lines 38-40: “The smaller and intrinsically labile d3mA and d7mG bases are conversely believed to be removed by hydrolysis due to increased lifetime of these bases in a solvent-exposed orientation (4).” This hypothesis was disproven by later papers from the same lab, and thus this sentence should be replaced with, “The smaller and intrinsically labile d3mA and d7mG bases are conversely believed to be removed by hydrolysis due to stabilization of the increased positive charge on the deoxyribose backbone through electrostatic and CH-pi interactions (Mullins et al, Nature (2015) and Parsons et al, JACS (2016) 138, 11485–11488).”

Addressed in lines 46-49

3. Line 36: N3-yatakemycinyldeoxyadenosine (d3yA) should be N3-yatakemycinyl-2'-deoxyadenosine (d3yA) to be consistent with the d3mA and d7mG nomenclature used.

Addressed in line 44.

4. Line 161, why was *E. faecalis* AlkD used rather than *B. cereus* AlkD, which has been used in most previous studies?

In an attempt to crystalize AlkD a few years ago we managed to get data for *E. faecalis* AlkD and not for *B. cereus* AlkD. This made *E. faecalis* AlkD a preferred choice in our lab for further experiments.

Reviewers' Comments:

Reviewer #1:

Remarks to the Author:

The reviewers addressed all of my major comments, and I feel the manuscript is ready for publication with the following minor comments:

Figure 1: This is semantics, but I believe Figure 1 shows a position vs. time plot, not a kymograph. Most kymographs are raw image projections, not position fitted data.

Figure 2: I greatly appreciated the reviewers' direct response to my question about the two diffusion modes of AlkD, and I now feel the body of the manuscript does an excellent job of explaining why AlkD lacks an interrogation mode. However, I am still struggling with seeing the single search mode for AlkD in Figure 2. In considering the slower mode of diffusion, AlkD and AlkF have very similar diffusion rates. One could argue that if AlkF has a slow interrogation mode with a diffusion constant $< 0.05 \text{ } \mu\text{m}^2/\text{s}$, then AlkD has that same mode. The difference between AlkD and AlkF really shows up in the fast mode of diffusion. AlkF is twice as fast as AlkD, and for this reason has a larger energy barrier difference between the two states and is considered "bimodal." In the Figure 2d-e caption, I think the terms "fast" and "slow" mode still lead the reader to conclude that AlkD has two distinct search modes that correlate directly to the modes in AlkF. Also, the fact that the colors directly correspond between the first and second panels of Fig. 2 encourages the conclusion that bimodal search behavior is similar between the homologs. Because the authors use the barrier difference to distinguish bimodal from single mode, it is my opinion that Fig. 2 would be strengthened if the authors better emphasized the barrier differences for the search modes of AlkD and AlkF.

Reviewer #2:

Remarks to the Author:

In the manuscript, the authors characterize at the single-molecule level the diffusive behaviour of bacterial glycosylases which belong to the HEAT-like repeat (HLR) family. Contrary to other classes of glycosylases, HLR family doesn't require a flipped nucleobase in the aberrant nucleotide (forming so-called "lesion-recognition pocket") to execute its activity. It is then hypothesised that lack of this requirement influences the lesion search mechanism and the presented single-molecule fluorescence data shows that AlkD and AlkF glycosylase scan the stretched DNA without a stable interrogation complex.

The paper analyses the "instantaneous diffusion rate" to detect "statistically significant variations in the diffusion rate of the proteins under study". This involves measuring local MSDs for a moving window of 5 frames and calculating the corresponding diffusion coefficient. It's an interesting approach that allows capturing slow and fast components of the process, provided that the distribution of the rates deviates from a random walk.

The resulting analysis reveals a bimodal distribution of diffusion rates of AlkF (that has a positively charged beta-hairpin) changes to mono-modal once positively-charged residues are mutated (neutralized) in the protein. AlkF-mutant scanning becomes "more homogenous and less-interrupted". The increase in the average diffusion rate, as well as the loss of bimodality, is correlated with the structural changes in the protein and its interaction with the DNA (e.g hopping mechanism that was not detected for AlkD and AlkF-WT within the resolution limit). These studies should be of interest to a broad community of single-molecule scientists and molecular biologists. However, I have some concerns on the rather subtle difference between monomodal and bimodal distribution in WT AlkD and AlkF scanning rate, respectively and I wonder how certain technical factors could influence the conclusions:

- The deviation from a random walk model is observed for AlkF (Fig 2d middle) but not for AlkD (Fig 2d left). The authors interpret the latter AlkD data as "the absence of an explicit bimodality" but one could argue that the AlkD distribution can be fitted with more than 1 Gaussian. Later on, the observation was supported with an independent HMM analysis that resulted in an energy barrier of 1.46 kBT which was assigned as too low for the existence of 2 stable states. This was further confronted with 2.3 kBT energy barrier measured for other proteins with explicit peaks in the diffusion. The same concern comes up – is this ~1 kBT difference big enough to say that AlkD doesn't have a stable intermediate whereas other proteins do?
- What would be an "artefactual" diffusion rate for a protein that is stably bound to the DNA? I guess MSDs from 5-frame moving windows would still give some small average displacement caused by the DNA thermal fluctuations and some tracking inaccuracies. Did the authors take this into consideration? What is the spatial precision here? Control experiments with a non-diffusive protein seem to be lacking.
- The authors assure in line 365 "By changing the width of the window to 7, 10 and 15 frames, we verified that the main features of the distribution of the instantaneous diffusion rates are independent of the width of the window". Why didn't they plot MSD as a function of the time window as it is usually done for this type of analysis?
- The authors never specify the force exerted on the DNA, or how this might affect the observed diffusion patterns. The authors need to specify the forces they are using in their experiments and report diffusion rates at multiple forces.
- Line 404/405 typo: "kinatic calssification"

Reviewer #3:

Remarks to the Author:

The authors have addressed most of my previous concerns. However, the authors have pivoted based on the reviews and changed their interpretation to focus more on a lack of an interrogation complex rather than the monomodal nature of scanning. I think this is an interesting and worthwhile study that adds to our understanding of how DNA repair enzymes scan DNA. However, while the data and the comparison to other glycosylases is consistent with a lack of an interrogation complex, I was not convinced that this can be firmly concluded by these data. That is, the data and its analysis suggests scanning without a stable interrogation complex. This is just a matter of toning down the language.

Minor point:

Line 116, "Fig 2b" should be "Fig 2d"

Reviewers' comments:

Reviewer #1 (Remarks to the Author):

The reviewers addressed all of my major comments, and I feel the manuscript is ready for publication with the following minor comments:

Figure 1: This is semantics, but I believe Figure 1 shows a position vs. time plot, not a kymograph. Most kymographs are raw image projections, not position fitted data.

We have changed the wording in the figure caption in line 84-85

Figure 2: I greatly appreciated the reviewers' direct response to my question about the two diffusion modes of AlkD, and I now feel the body of the manuscript does an excellent job of explaining why AlkD lacks an interrogation mode. However, I am still struggling with seeing the single search mode for AlkD in Figure 2. In considering the slower mode of diffusion, AlkD and AlkF have very similar diffusion rates. One could argue that if AlkF has a slow interrogation mode with a diffusion constant $< 0.05 \text{ } \mu\text{m}^2/\text{s}$, then AlkD has that same mode. The difference between AlkD and AlkF really shows up in the fast mode of diffusion. AlkF is twice as fast as AlkD, and for this reason has a larger energy barrier difference between the two states and is considered "bimodal." In the Figure 2d-e caption, I think the terms "fast" and "slow" mode still lead the reader to conclude that AlkD has two distinct search modes that correlate directly to the modes in AlkF. Also, the fact that the colors directly correspond between the first and second panels of Fig. 2 encourages the conclusion that bimodal search behavior is similar between the homologs. Because the authors use the barrier difference to distinguish bimodal from single mode, it is my opinion that Fig. 2 would be strengthened if the authors better emphasized the barrier differences for the search modes of AlkD and AlkF.

We agree with the reviewer in emphasizing the barrier difference for the Markov-classified modes of AlkD and AlkF, which provides a more elaborate explanation of the observed contrasts between the scanning by these proteins. In addition, when it comes to formation of an interrogation complex, it is important that we compare AlkD to the proteins that are known to form interrogation complexes. For this purpose, based on the two separate peaks in the overall diffusion rate distribution of OGG1 and EndoV from our previous study¹, we had estimated that energy barrier difference to be minimum $2.3 k_B T$. In this round of revision, we calculated the accurate energy barrier difference between the two Markov-classified modes of OGG1 and EndoV to be $2.66 k_B T$ and $2.30 k_B T$, respectively. Both of these proteins are known to form stable interrogation complexes, making the comparison with AlkD relevant. Therefore, we visualize the energy barrier differences between the two modes of OGG1 and EndoV along with those of the corresponding data for AlkD and AlkF in an inset plot in fig. 2b. In this plot, we have also indicated the estimated range of energy barrier variation for helical sliding ($2k_B T - 0.5k_B T = 1.5 k_B T$). We see that the difference in energy barrier of the two modes of AlkD still lies within the helical sliding range, while for hOGG1, EndoV and AlkF the difference clearly exceeds this range, indicating that at least one of the modes for these proteins is distinct from helical sliding. In response to this point, we added in inset to fig. 2d and implemented changes in line 106-127, 163-165, 173-174

In addition, we added a sentence in fig. 2 caption (line 158-160) and the method part (429-432), clarifying that the two classified modes for each protein do not necessarily correlate with those of other proteins.

Reviewer #2 (Remarks to the Author):

In the manuscript, the authors characterize at the single-molecule level the diffusive behaviour of bacterial glycosylases which belong to the HEAT-like repeat (HLR) family. Contrary to other classes of glycosylases, HLR family doesn't require a flipped nucleobase in the aberrant nucleotide (forming so-called "lesion-recognition pocket") to execute its activity. It is then hypothesised that lack of this requirement influences the lesion search mechanism and the presented single-molecule fluorescence data shows that AlkD and AlkF glycosylase scan the stretched DNA without a stable interrogation complex.

The paper analyses the "instantaneous diffusion rate" to detect "statistically significant variations in the diffusion rate of the proteins under study". This involves measuring local MSDs for a moving window of 5 frames and calculating the corresponding diffusion coefficient. It's an interesting approach that allows capturing slow and fast components of the process, provided that the distribution of the rates deviates from a random walk.

The resulting analysis reveals a bimodal distribution of diffusion rates of AlkF (that has a positively charged beta-hairpin) changes to mono-modal once positively-charged residues are mutated (neutralized) in the protein. AlkF-mutant scanning becomes "more homogenous and less-interrupted". The increase in the average diffusion rate, as well as the loss of bimodality, is correlated with the structural changes in the protein and its interaction with the DNA (e.g hopping mechanism that was not detected for AlkD and AlkF-WT within the resolution limit). These studies should be of interest to a broad community of single-molecule scientists and molecular biologists. However, I have some concerns on the rather subtle difference between monomodal and bimodal distribution in WT AlkD and AlkF scanning rate, respectively and I wonder how certain technical factors could influence the conclusions:

- The deviation from a random walk model is observed for AlkF (Fig 2d middle) but not for AlkD (Fig 2d left). The authors interpret the latter AlkD data as "the absence of an explicit bimodality" but one could argue that the AlkD distribution can be fitted with more than 1 Gaussian. Later on, the observation was supported with an independent HMM analysis that resulted in an energy barrier of 1.46 kBT which was assigned as too low for the existence of 2 stable states. This was further confronted with 2.3 kbT energy barrier measured for other proteins with explicit peaks in the diffusion. The same concern comes up – is this ~1 kbT difference big enough to say that AlkD doesn't have a stable intermediate whereas other proteins do?

As shown in the figure below (and the red curve in Fig. 2d), the absence of an explicit bimodality in the overall diffusion rate distribution of AlkD and its contrast with other proteins, including AlkF as well as OGG1 and EndoV from our previous study ¹, is a meaningful observation, especially since these plots represent the collective behavior of hundreds of scanning trajectories for each proteins.

Moreover, we show that although the scanning of AlkD is classified into two modes a contrast between AlkD on one hand, and the proteins known to form interrogation complexes on the other hand (Ogg1 and EndoV), is evident by looking at the energy barrier difference of the two classified modes (new inset in Fig 2d).

The energy barrier of scanning is a variable with a rather narrow range of variation. As suggested by a hallmark theoretical study², the energy barrier of scanning for efficient helical sliding is around $1 k_B T$, and at around $2 k_B T$ scanning becomes extremely slow and interrogation complexes may form. Therefore a $1 k_B T$ difference in energy barrier of scanning can define the mode with which the protein interacts with the DNA. Based on these theoretical predications^{2,3} and several experimental reports^{1,4-9} the range of energy barrier variation for helical sliding is $2k_B T - 0.5k_B T = 1.5 k_B T$.

Moreover, the mentioned $2.3 k_B T$ difference for other proteins was a conservative estimate based on the diffusion rate difference of the two explicit peaks as seen in the diffusion rate distribution. Using the data of our previous study, we performed a 2-mode Markov classification and calculated the accurate energy barrier difference between these two modes for hOGG1 and EndoV to be $2.66 k_B T$ and $2.30 k_B T$, which exceeds the $1.5k_B T$ range of helical sliding by 77% and 53%, respectively. We have included an inset in Fig. 2d illustrating the energy barrier differences for OGG1, EndoV, AlkD and AlkF, along with the range of energy barrier variation for helical sliding. To further clarify these points, we have included some changes in the manuscript in line 106-127, 163-165, 173-174

- What would be an “artefactual” diffusion rate for a protein that is stably bound to the DNA? I guess MSDs from 5-frame moving windows would still give some small average displacement caused by the DNA thermal fluctuations and some tracking inaccuracies. Did the authors take this into consideration? What is the spatial precision here? Control experiments with a non-diffusive protein seem to be lacking.

The average diffusion rate of 67 protein trajectories (for AlkD/AlkF) with confined movement on DNA is calculated to be $0.0086 \pm 0.0084 \mu\text{m}^2/\text{s}$. This value of diffusion rate can have contributions from DNA thermal fluctuation as well as tracking inaccuracies. In addition, due to intrinsic resolution limitation of our single-molecule fluorescence microscopy approach, we are not able to confirm whether proteins were stably bound to a specific base on DNA or they move between adjacent bases with a rate and resolution undetectable by our system. Therefore, it would be difficult to estimate the effect of thermal fluctuation on the calculated diffusion rate directly and isolated from other factors. However, we have calculated the contribution of tracking inaccuracies to the diffusion rate by looking at 52 trajectories where proteins are stably bound to the coverslip surface. The diffusion rate was calculated to be $0.0066 \pm 0.0060 \mu\text{m}^2/\text{s}$ which is well below the resolved features of

diffusion rate distributions and average diffusion rates of the Markov states. In addition, from these data we have calculated the spatial precision to be 18.7 ± 3.0 nm. We have included this information in lines 363-365.

- The authors assure in line 365 “By changing the width of the window to 7, 10 and 15 frames, we verified that the main features of the distribution of the instantaneous diffusion rates are independent of the width of the window”. Why didn’t they plot MSD as a function of the time window as it is usually done for this type of analysis?

The width of this moving window defines how many consecutive frames are considered in the calculation of the instantaneous diffusion rate for any given frame within each trajectory, thus determining the sensitivity of the calculated diffusion rate to instantaneous variations. By changing the width of the moving window we confirm that the overall shape of diffusion rate distribution (the red curve in fig. 2d) remains consistent using larger moving windows as well. Using the concept of instantaneous diffusion rate, we are able to look at the distribution of diffusion rate which in turn allows for collective detection of variations in diffusion rate. For example, as shown in the figure presented in response to the first remark by the reviewer, the variations in the diffusion rate is detected as explicit bimodalities in the overall diffusion rate distribution for proteins other than AlkD. Such variations remain undetectable in a conventional MSD – time interval plot used in previous studies^{5,8,9}. As an example, in our previous study we detected the modality of scanning by OGG1 using the concept of instantaneous diffusion rate¹ while such behavior was undetected in a previous study using conventional diffusion analysis on the same protein⁵. Therefore, instantaneous diffusion rate is generally recommended for an unbiased analysis of a movement with variable diffusion rate¹⁰, and we have shown how this allows for detection of variations in the diffusion rate which in turn represent different scanning modes.

- The authors never specify the force exerted on the DNA, or how this might affect the observed diffusion patterns. The authors need to specify the forces they are using in their experiments and report diffusion rates at multiple forces.

Our current setup does not allow for a direct determination of the exerted force. However, by keeping the power of the laser and the displacement of the bead from the trap center consistent, we ensured that the exerted force remained constant throughout all experiments. This contrasts with previous experiments by others using DNA tigtropes or flow to elongate the DNA, where the tension in the DNA is not under control. The exerted force in our experiments led to elongation of DNA to around 95 % of its theoretical contour length, which, according to universal force extension curves for double stranded DNA, corresponds to forces of around 10 pN. More importantly, the factor of force remains constant in all experiments and cannot have affected the observed diffusion patterns and distributions of the proteins under study.

We have included clarification related to the exerted forces on DNA in line 327-329.

- Line 404/405 typo: “kinatic calssification”

We have fixed this typo in line 417.

Reviewer #3 (Remarks to the Author):

The authors have addressed most of my previous concerns. However, the authors have pivoted based on the reviews and changed their interpretation to focus more on a lack of an interrogation

complex rather than the monomodal nature of scanning. I think this is an interesting and worthwhile study that adds to our understanding of how DNA repair enzymes scan DNA. However, while the data and the comparison to other glycosylases is consistent with a lack of an interrogation complex, I was not convinced that this can be firmly concluded by these data. That is, the data and its analysis suggests scanning without a stable interrogation complex. This is just a matter of toning down the language.

In the revised version of the manuscript, we have further tried to clarify our argument for lack of an interrogation complex by focusing on the energy barrier difference calculated from the diffusion rates of the two different states predicted by the Markov analysis. With the new inset in Fig 2d, we show that the difference between the two modes of scanning for AlkD does not exceed the range expected for normal helical sliding, as reported through both theoretical and experimental studies. This is in contrast with proteins known to form interrogation complexes since the energy barrier difference between the two modes exceeds well beyond this range for those proteins.

The relevant applied changes can be found in fig. 2d and line 106-127

Minor point:

Line 116, "Fig 2b" should be "Fig 2d"

We have fixed this in line 118.

1. Ahmadi, A. *et al.* Breaking the speed limit with multimode fast scanning of DNA by Endonuclease V. *Nat. Commun.* **9**, 5381 (2018).
2. Slutsky, M. & Mirny, L. A. Kinetics of Protein-DNA Interaction: Facilitated Target Location in Sequence-Dependent Potential. *Biophys. J.* **87**, 4021–4035 (2004).
3. Bagchi, B., Blainey, P. C. & Sunney Xie, X. Diffusion constant of a nonspecifically bound protein undergoing curvilinear motion along DNA. *J. Phys. Chem. B* **112**, 6282–6284 (2008).
4. Dunn, A. R., Kad, N. M., Nelson, S. R., Warshaw, D. M. & Wallace, S. S. Single Qdot-labeled glycosylase molecules use a wedge amino acid to probe for lesions while scanning along DNA. *Nucleic Acids Res.* **39**, 7487–7498 (2011).
5. Blainey, P. C., van Oijen, A. M., Banerjee, A., Verdine, G. L. & Xie, X. S. A base-excision DNA-repair protein finds intrahelical lesion bases by fast sliding in contact with DNA. *Proc. Natl. Acad. Sci. U. S. A.* **103**, 5752–5757 (2006).
6. Tafvizi, A. *et al.* Tumor suppressor p53 slides on DNA with low friction and high stability. *Biophys. J.* **95**, L01–L03 (2008).
7. Blainey, P. C. *et al.* Nonspecifically bound proteins spin while diffusing along DNA. *Nat. Struct. Mol. Biol.* **16**, 1224–1229 (2009).
8. Tafvizi, A., Huang, F., Fersht, A. R., Mirny, L. a & van Oijen, A. M. A single-molecule characterization of p53 search on DNA. *Proc. Natl. Acad. Sci. U. S. A.* **108**, 563–8 (2011).
9. Bonnet, I. *et al.* Sliding and jumping of single EcoRV restriction enzymes on non-cognate DNA. *Nucleic Acids Res.* **36**, 4118–4127 (2008).

10. Lin, G. G. & Scott, J. G. Mean Square Displacement Analysis of Single-Particle Trajectories with Localization Error: Isotropic Brownian Motion in Medium. **100**, 130–134 (2012).